# Learning from Concealed Labels

## ABSTRACT

Annotating data for sensitive labels (e.g., disease, smoking) poses a potential threats to individual privacy in many real-world scenarios. To cope with this problem, we propose a novel setting to protect privacy of each instance, namely learning from concealed labels for multi-class classification. Concealed labels prevent sensitive labels from appearing in the label set during the label collection stage, as shown in Figure 1, which specifies none and some random sampled insensitive labels as concealed labels set to annotate sensitive data. In this paper, an unbiased estimator can be established from concealed data under mild assumptions, and the learned multi-class classifier can not only classify the instance from insensitive labels accurately but also recognize the instance from the sensitive labels. Moreover, we bound the estimation error and show that the multi-class classifier achieves the optimal parametric convergence rate. Experiments demonstrate the significance and effectiveness of the proposed method for concealed labels in synthetic and real-world datasets.

## CCS CONCEPTS

• **Computing methodologies** → *Supervised learning by classification.*

## KEYWORDS

Concealed labels, Weakly supervised learning, Unbiased estimator, Privacy labels learning, Corrected risk estimator

## 1 INTRODUCTION

Traditional ordinal supervised learning tasks face many challenges, where obtaining massive amounts of data with accurate supervised information is difficult, nay impossible in some real-world scenarios. To mitigate this problem, various weakly supervised learning frameworks [2, 16, 25, 29] have been extensively studied to bring a new inspiration for improving learning performance, including semi-supervised learning [8, 17, 18, 34], positive-unlabeled learning [4, 5, 11, 12, 27], multi-instance learning [19, 20, 24, 42] and noisy-label learning [21, 31, 36, 37].

Another critical challenge in obtaining a large number of high-quality labels arises when sensitive information cannot be released to public [14, 15, 28]. For example, in both business and personal life, there is a wealth of sensitive information (e.g., political preferences or habits), whose labeling information needs to be concealed during data collection. In this problem, collecting explicit

*ACM MM, 2024, Melbourne, Australia*

© 2024 Copyright held by the owner/author(s). Publication rights licensed to ACM.
ACM ISBN 978-x-xxxx-xxxx-x/YY/MM
https://doi.org/10.1145/nnnnnnn.nnnnnnn

sensitive labels becomes difficult, prohibiting learning from ordinal supervised data. To overcome this bottleneck, researchers have explored privacy-label learning approaches, such as Label Proportions Learning (LPL)[1, 6, 23, 40], Complementary Label Learning (CoLL) [13, 14, 22], pairwise similar learning [7, 10, 29], etc. LPL is a well-studied setting that protects the sensitive information by annotating the proportions of positive instances instead of providing explicit labels. CoLL is another widely used weakly supervised learning method that protects privacy by specifying one of the labels which the instance does not belong. Although those weakly supervised learning approaches can protect sensitive information during the label collection stage, they conceal all labels for each instance, regardless of weather or not the label contains sensitive information. Therefore, due to the complete absence of precisely labeled data, these methods increase the difficulty of training the classifier.

In this paper, we consider a novel privacy protection weakly supervised learning setting, aiming to conceal sensitive labels during the annotation of instances. Under this setting, as shown in Figure 1, the Concealed Label (CL) is introduced to prevent sensitive labels from appearing in the label set during the label collection stage. CL specifies none label and some random sampled insensitive labels as concealed labels set to annotate sensitive data. This concealed labels setting existing in many real-world scenarios. For example, some individuals hesitate to admit to smoking in their daily life, but they are willing to share information about cycling, drinking and phoning, as shown in Figure 1. Another example is annotating instances related to disease, where patients with diseases require more privacy protection for their data compared to normal individuals. Therefore, when we consider data privacy, sensitive labels will not appearing in the label set. Fortunately, we can collect concealed labels for data to train a multi-class classifier.

The goal of this paper is to propose a novel framework for learning from concealed labels, which utilizes the none label for ensuring that the sensitive label is not disclosed to adversary. It is important to note that the learned classifier has the capability to accurately recognize instances from unconcealed labels and identify instances from concealed labels. The contributions of this paper can be summarized as follows:

(1) We propose a novel privacy-label weakly supervised learning setting, i.e., learning from concealed labels, which prevents sensitive labels from appearing in the label set.
(2) We propose an empirical risk minimization method that constructs an unbiased estimator for multi-class classification using concealed labels data, and provides estimation error bounds for the proposed method.
(3) We experimentally demonstrate that the learned classifier is useful for recognizing instances from both unconcealed and conceal labels on various benchmark datasets and two real-world concealed labels datasets.

The rest of this paper is structured as follows. Section 2 reports on related work. Section 3 gives formal definitions about conceal

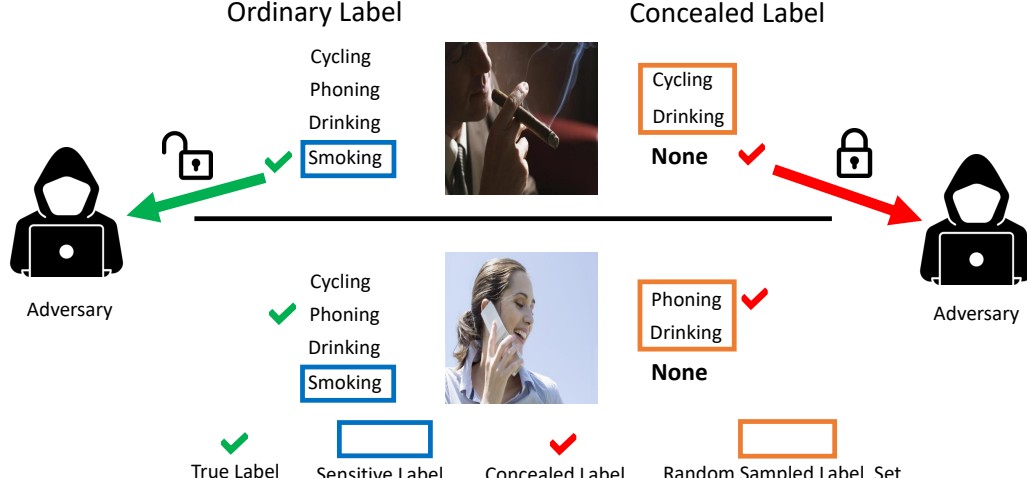

**Figure 1: Illustrations a example of concealed labels during the annotation procedure in real-world scenario. Smoking, being a sensitive label, is often a challenging attribute to collect data, due to people's hesitancy in admitting their smoking habits in daily life. To ensure privacy protection, it is crucial not to include the sensitive label. Concealed labels are employed to prevent the inclusion of the sensitive label that needs to be concealed. By utilizing the none label, data privacy can be safeguarded, ensuring that the sensitive label remains undisclosed for adversary.**

labels and presents the proposed approach with theoretical analyses. Section 4 reports the results of the comparative experiment. Finally, Section 5 concludes this paper.

## 2 RELATED WORK

In this section, we discuss several research topics that are related to concealed labels, including privacy-label learning and positive and unlabeled learning.

### 2.1 Privacy-label learning

Recently, to protect the privacy of the data during instance annotating, researchers have studied various privacy label annotating settings, including Label Proportions Learning (LPL) [9, 28, 40], Complementary Label Learning (CoLL) [14], Similarity and Unlabeled Learning (SUL) [10, 29] and Similarity-Confidence Learning (SCL) [7, 43].

LPL [9, 28, 40] aims to protect sensitive information by annotating the proportions of positive instances in the bag instead of specifying the label directly. LPL can be used in the field of medical and health classification, such as disease prediction, where the patient pay attention to privacy information. Given the difficulty in acquiring fully supervised data, the adoption of the LPL setting emerges as a comparatively secure approach to safeguarding privacy.

SUL [10, 29] and SCL [7, 43] aim to train a binary classifier using only unlabeled data pairs, where data annotation is based on the similarity between two instances rather than explicit label assignment for each instance to protect privacy. In this scenario, individuals are unable to precisely discern the sensitive label. If the instance pair is annotated with a similar label, this helps prevent substantial privacy leakage.

In addition, CoLL [14] and its cousin, i.e., Partial Labels Learning (PLL) [3, 32, 35, 39] have tried to addressed the privacy protection problem by specifying label that the instance does not belong to. For example, collecting some medical data may require privacy questions, which would be more mentally demanding. It would be easy for the patient to provide some incorrect answers rather than sensitive exactly label, which is a safe way for collecting privacy labels.

To best of our knowledge, previous privacy-label learning approaches conceal all labels for each instance, even if some labels does not contain sensitive information. Therefore, those approaches increase the difficulty for training the classifier. In this paper, we propose the concealed labels learning framework to overcome this limitation, reducing the overhead of learning from insensitive labels on real-world datasets.

### 2.2 Positive and unlabeled learning

Another line of related works focuses on a widely studied weakly supervised framework called Positive and Unlabeled Learning (PUL) [4, 11, 33, 45], which trains a binary classifier using only positive and unlabeled data without negative instance. PUL is a special semi-supervised learning task which can be used for augmented classes learning [44]. Due to the absence of labeled negative instances, PUL provides privacy protection for the negative label in data collection tasks. However, while PUL effectively leverages the unknown negative instances in the unlabeled dataset, it does not offer privacy protection for the sensitive information associated with positive labels, which are more important in real-world tasks.

Recently, [38] extended PUL to handle multi-class data (MPUL) [30]. This approach utilizes labeled instances from multiple positive labels, and unlabeled instances from the unknown negative label to

learn a multi-class classifier. However, the MPUL methods can not actively protect the sensitive label in annotation process, which is also important in real-world tasks. Specifically, to protect privacy information, the MPUL methods must collect two datasets, one labeled dataset (including sensitive label) and one unlabeled dataset. Then, the already labeled data (with sensitive label) would be set to unlabeled data, which exposes sensitive labels to adversaries. In contrast, conceal labels prevent the inclusion of the sensitive label, ensuring that the sensitive label remains undisclosed to adversaries, as shown in Figure 1 in the main text.

## 3 METHODOLOGY

In this section, we present a formal description of learning from concealed labels, focusing on the construction of an unbiased risk estimator using the concealed data distribution. Besides, we propose a correction risk estimator to enforce the risk to be non-negative. Furthermore, we prove the estimation error bound of proposed unbiased risk estimator.

### 3.1 Problem setup

Since the specific partition of labels that need to be concealed is unknown, the classifier will recognize all of them as a single label $cl$. Let $\mathcal{X} \subset R^d$ and $\mathcal{Y} = \{1, ..., K, cl\}$ be a d-dimensional instance and multi-class label space, where $K$ denotes the number of classes, respectively, and let $D = \{(x_i, Y_i, s_i)\}_{i=1}^n$ be concealed labels data sampled from distribution $P(x, Y, s)$ defined over $\mathcal{X} \times \mathcal{Y}_r \times \mathcal{S}$, where $\mathcal{Y}_r \subset \{1, ..., K\}$ is random sampled label set space, $Y_i \in \mathcal{Y}_r$ is a set of random sampled labels for instance $x_i \in \mathcal{X}$, and $s_i$ is sampled from concealed labels space $\mathcal{S} = \{1, ..., K, s_{none}\}$ for instance $x_i$, and $s_{none}$ denotes the label of none. We denote the event that the true label $y$ of instance $x$ does not appear in random sampled label set $Y$ (i.e., $y \notin Y$) by $s = s_{none}$, and otherwise by $s = y$.

In our setup, the randomly sampled label set $Y$ contains $L$ labels, and the probability of each label appearing in this set is the same, denoted as $P(y_k \in Y) = P(y_{k'} \in Y)$, where $y_k$ and $y_{k'}$ represent two sampled labels from the random sampled label set. Additionally, the randomly sampled label set offers some weak supervisory information for concealed label data by excluding certain incorrect labels.

It is worth noting that $s_{none}$ and sensitive labels are not equivalent. While all sensitive labels are annotated as $s_{none}$, some non-sensitive labels can also be annotated as $s_{none}$. Next, to formulate the generation process of concealed labels data and derive unbiased risk estimation, we introduce the following assumption, which facilitates the implementation of our approach for real-world data annotation tasks.

**Definition 1** (Concealed Labels Assumption). The conditional distribution of concealed labels, i.e., $\{P(s = s_{none}|x, y = i)\}_{i=1}^K \cup P(s = s_{none}|x, y = cl)$ and $P(s = j \neq \{i \wedge s_{none}\}|x, y = i)$ are under the concealed labels assumption as follows:

$$P(s = s_{none}|x, y = cl) = 1 \tag{1}$$

$$P(s = j \neq \{i \wedge s_{none}\}|x, y = i) = 0 \tag{2}$$

$$P(s = s_{none}|x, y = 1) = P(s = s_{none}|x, y = 2)$$
$$\vdots \tag{3}$$
$$= P(s = s_{none}|x, y = K)$$
$$= (K - L)/K$$

Concealed labels assumption states that the data sampled from unconcealed labels is annotated as $s = s_{none}$ with uniform probability. Then, the random sampled label set $Y$ can be generated easily by sampling labels from unconcealed labels set $\mathcal{Y}_u = \{1, ..., K\}$ with uniform probability.

The goal of learning from concealed labels is to obtain a multi-class classifier $f : \mathcal{X} \to \mathcal{Y}_m$ by minimizing the expected ordinary supervised risk as follows:

$$R_m(f) = \mathbb{E}_{(x,y)\sim p(x,y)} \mathcal{L}(f(x), y)$$
$$= \mathbb{E}_{x\sim M} \left[ \left[ \sum_{i=1}^K p(y = i|x) \mathcal{L}(f(x), i) \right] \right.$$
$$\left. + p(y = cl|x) \mathcal{L}(f(x), cl) \right] \tag{4}$$

where $M := P(x)$, $P(x, y)$ and $P(y = i|x)$ denote the joint and conditional distributions of ordinary supervised data and $\mathcal{L}(f(x), y)$ denotes the multi-class loss function.

### 3.2 Unbiased risk estimator

In this section, we present our formulation of unbiased risk for learning the multi-class classifier using only concealed labels data, based on the setup described above. In Eq.(4), the conditional distribution $P(y = i|x)$ is unavailable for training the multi-class classifier since we do not have access to ordinary supervised data. Fortunately, we can use concealed labels data to represent it by introducing the conceal labels conditional distribution $P(s = i|x)$ and $P(s = s_{none}|x)$.

**Lemma 2.** Under the concealed labels assumption, we can express conditional distribution $P(y = i \neq cl|x)$ and $P(y = cl|x)$ in terms of $P(s = i|x)$, $P(s = s_{none}|x)$ as

$$P(y = i \neq cl|x) = \frac{K}{L} P(s = i|x) \tag{5}$$

$$P(y = cl|x) = \frac{K}{L} P(s = s_{none}|x) - \frac{K - L}{L} \tag{6}$$

**Proof Sketch.** To prove the conditional distribution, we rewrite the probability $P(s = y_i|x)$ according to the Definition 1, and prove $P(y = i \neq cl | x) = P(s = i | x) + \frac{K-L}{K} P(y = i \neq cl | x)$. Then, we prove $P(y = cl | x) = \sum_Y P(Y, s = s_{none}, y = cl | x)$ in a similar way and decompose the probability $P(s = s_{none}, Y | x)$ into four parts. Finally, we demonstrate $P(y = cl|x) = P(s = s_{none} | x) + \frac{K-L}{K} p(y = cl | x) - \frac{K-L}{K}$ by substituting the rewritten probability $P(s = s_{none}, Y | x)$ into $\sum_Y P(Y, s = s_{none}, y = cl | x)$ and using the Definition 1. □

The main technique for proving this lemma is to utilize the Bayes Rule and Total Probability Theorem, then we can prove this lemma. The more details of the proof is provided in Appendix.

Thus, By plugging Eq.(5) and (6) into Eq.(4), we can evaluate the ordinary supervised classification risk $R_m(f)$ using an equivalent risk $R_{CL}(f)$ during the training stage.

**Theorem 3.** Under the concealed labels assumption, for multi-class classifier $f$, we have $R_m(f) = R_{CL}(f)$, where $R_{CL}(f)$ is defined as

$$R_{CL}(f) = \mathbb{E}_{(x,s)\sim P(x,s\neq s_{none})} \frac{K}{L} \mathcal{L}(f(x),s)$$
$$+ \mathbb{E}_{(x,s)\sim P(x,s=s_{none})} \frac{K}{L} \mathcal{L}(f(x),cl) \qquad (7)$$
$$- \mathbb{E}_M \frac{K-L}{L} \mathcal{L}(f(x),cl)$$

*Proof.* According to the Lemma 2, we have

$$R_m(f)$$
$$= \mathbb{E}_{x\sim M} \left\{ \sum_{i=1}^{K} P(y=i\mid x)\mathcal{L}(f(x),i) \right.$$
$$\left. + P(y=cl\mid x)\mathcal{L}(f(x),cl) \right\}$$
$$= \mathbb{E}_{x\sim M} \sum_{i=1}^{K} \frac{K}{L}P(s=i\mid x)+ \qquad (8)$$
$$\left[ \frac{K}{L}P(s=s_{none}\mid x) - \frac{K-L}{L}\mathcal{L}(f(x),cl) \right]$$
$$= \mathbb{E}_{(x,s)\sim P(s,s\neq s_{none})} \frac{K}{L}\mathcal{L}(f(x),s)$$
$$+ \mathbb{E}_{(x,s)\sim P(s,s=s_{none})} \frac{K}{L}\mathcal{L}(f(x),cl)$$
$$- \mathbb{E}_M \frac{K-L}{L}\mathcal{L}(f(x),cl)$$
$$= R_{CL}(f)$$

which concludes the proof. $\square$

As we can see from Eq.(7), $R_{CL}(f)$ can be assessed in the training stage using the number of classes $K$ and sampled labels set $L$.

Here, we rearrange our concealed labels dataset as $\mathcal{X}_c = \{\mathcal{X}_s\}_{s=1}^{K} \cup \mathcal{X}_{none}$, where $\mathcal{X}_s$ and $\mathcal{X}_{none}$ denote the samples with concealed labels $s \neq s_{none}$ and $s = s_{none}$ respectively. Then, the classification risk $R_{CL}$ can be approximated by

$$\widehat{R}_{CL}(f) = \frac{1}{\#\{\mathcal{X}_s\}_{s=1}^{K}} \sum_{s=1}^{K} \sum_{x_j\in\mathcal{X}_s} \frac{K}{L}\mathcal{L}(f(x_j),s)$$
$$+ \frac{1}{\#\mathcal{X}_{none}} \sum_{x_j\in\mathcal{X}_{none}} \frac{K}{L}\mathcal{L}(f(x_j),cl) \qquad (9)$$
$$- \frac{1}{\#\mathcal{X}_c} \sum_{x_j\in\mathcal{X}_c} \frac{K-L}{L}\mathcal{L}(f(x_j),cl)$$

where $\#\mathcal{X}_i$ denotes the number of samples in the data set $\mathcal{X}_i$. We can then train a multi-class classifier by minimizing the proposed empirical approximation of the unbiased risk estimator in Eq.(9).

### 3.3 Corrected risk estimator

Since the classification risk is an expectation over non-negative loss function $\mathcal{L}(f(x),y)$, both the risk and its empirical approximator

have lower bounds, i.e., $R_{CL}(f) \geq 0$ and $\hat{R}_{CL}(f) \geq 0$. In fact, similar to issue of the empirical approximator going negative in binary classification from positive and unlabeled data, Eq.(9) can also become negative due to the negative term, when a flexible model is used. Therefore, the proposed risk estimator suffers from overfitting during the training of the multi-class classifier.

Each term in the ordinary supervised risk is non-negative, indicating that the optimal risk corresponding to each label is also non-negative and approaches to zero. Therefore, we can reformulate Eq.(7) to express the counterpart risk for each label as follows:

$$R_{CL}(f) = \mathbb{E}_M \sum_{i=1}^{K} \left[ \overbrace{\frac{K}{L}P(s=i\mid x)}^{P(y=i\mid x)} \right] \mathcal{L}(f(x),i)$$
$$+ \mathbb{E}_M \left[ \underbrace{\frac{K}{L}P(s=s_{none}) - \frac{K-L}{L}}_{P(y=cl\mid x)} \right] \mathcal{L}(f(x),cl) \qquad (10)$$

Enforcing the classification risk to be non-negative is a useful approach in the context of weakly supervised learning, such as, binary classification from positive and unlabeled data and learning from complementary labels. Here, we propose a correction risk estimator for learning from concealed labels by

$$\widehat{R}_{CL}^{g}(f) = \frac{1}{\#\{\mathcal{X}_s\}_{s=1}^{K}} \sum_{s=1}^{K} \sum_{x_j\in\mathcal{X}_s} \frac{K}{L}\mathcal{L}(f(x_j),s)$$
$$+ g\left[ \frac{1}{\#\mathcal{X}_{none}} \sum_{x_j\in\mathcal{X}_{none}} \frac{K}{L}\mathcal{L}(f(x_j),cl) \right. \qquad (11)$$
$$\left. - \frac{1}{\#\mathcal{X}_c} \sum_{x_j\in\mathcal{X}_c} + \frac{K-L}{L}\mathcal{L}(f(x_j),cl) \right]$$

where $g[z]$ denotes the correction function, such as the max-operator function $g[z] = max\{0,z\}$.

Although the correction empirical risk using max-operator ensures non-negative for certain mini-batch, it prevents the risk of each label from approaching to zero. Instead, it neglects the optimization of negative risk, which cannot decrease the degree of overfitting. To address this issue, an alternative correction function $g[z] = |z|$ can be employed to alleviate overfitting. Here, $|z|$ denotes the absolute value of $z$, i.e., $|z| = max\{0,z\} - min\{0,z\}$. This correction function ensures that the risk of each label approaches zero during the training stage, making it a preferable choice to mitigate overfitting.

### 3.4 Practical Implementation

In this section, we introduce the practical implementation of the proposed method.

**Loss functions.** As discussed in the previous section, the classification risk of learning from concealed labels can be recovered using arbitrary loss functions. One common approach is to employ the One-Versus-Rest (OVR) [41] strategy, where the binary surrogate losses $\phi : R \rightarrow [0,+\infty)$ are utilized. Examples of such surrogate losses include the logistic loss $\phi(z) = log(1 + exp(-z))$, hinge loss

$\phi(z) = max\{0, 1 - z\}$ and Square Loss (SL) $\phi(z) = (1 - z)^2$. The OVR strategy has theoretical guarantees and demonstrates good practical performance in multi-class supervised learning scenarios.

Additionally, the popular softmax cross-entropy loss can be employed within the proposed method to learn a multi-class classifier. This loss function is widely used in deep learning approaches and offers effective training for multi-class classification tasks.

**Model.** Our method can be implemented using deep learning or other classifier, such as linear classifiers, etc. However, due to the large number of parameters in deep learning models, directly optimizing a deep model may lead to overfitting and negative risk. Then, we can utilize the correction methods proposed in section 3.3 to train deep models for learning from conceal labels.

## 3.5 Estimation error bound

Here, the estimation error bound of the proposed unbiased risk estimator is derived to theoretically justify the effectiveness of our approach when implemented using deep neural networks with OVR strategy. Let $\mathbf{f} = [f_1, ..., f_K, f_{cl}]$ denote the classification vector function in the hypothesis set $\mathcal{F}$. We assume that there exist a constant $C_\phi > 0$, such that $sup_z\phi(z) \leq C_\phi$. Let $L_\phi$ be the Lipschitz constant of $\phi$, we can introduce the following lemma.

**Lemma 4.** For any $\delta > 0$, with the probability at least $1 - \delta$,

$$\sup_{\mathbf{f}\in\mathcal{F}}\left|R_s(\mathbf{f}) - \widehat{R}_s(\mathbf{f})\right|$$
$$\leq 2L_\phi\mathfrak{R}_{n_s}(\mathcal{F}) + 2\frac{C_\phi K}{L}\sqrt{\frac{\ln(2/\delta)}{2n_s}} \tag{12}$$

$$\sup_{\mathbf{f}\in\mathcal{F}}\left|R_{none}(\mathbf{f}) - \widehat{R}_{none}(\mathbf{f})\right|$$
$$\leq 2L_\phi\mathfrak{R}_{n_{none}}(\mathcal{F}) + 2\frac{C_\phi(K-L)}{L}\sqrt{\frac{\ln(2/\delta)}{2n_{none}}} \tag{13}$$

$$\sup_{\mathbf{f}\in\mathcal{F}}\left|R_c(\mathbf{f}) - \widehat{R}_c(\mathbf{f})\right|$$
$$\leq 2L_\phi\mathfrak{R}_{n_c}(\mathcal{F}) + 2\frac{C_\phi K}{L}\sqrt{\frac{\ln(2/\delta)}{2n_c}} \tag{14}$$

where $R_s(\mathbf{f}) = \mathbb{E}_{(x,s)\sim P(x,s\neq s_{none})}\frac{K}{L}\mathcal{L}(f(x),s)$, $R_c(\mathbf{f}) = \mathbb{E}_M\frac{K-L}{L}\mathcal{L}(f(x),cl)$, $R_{none}(\mathbf{f}) = \mathbb{E}_{(x,s)\sim P(x,s=s_{none})}\frac{K}{L}\mathcal{L}(f(x),cl)$ and $\widehat{R}_s(\mathbf{f})$ denote the empirical risk estimator to $R_s(\mathbf{f})$, $R_{none}(\mathbf{f})$ and $R_c(\mathbf{f})$ respectively. $\mathfrak{R}_{n_s}(\mathcal{F})$, $\mathfrak{R}_{n_{none}}(\mathcal{F})$ and $\mathfrak{R}_{n_c}(\mathcal{F})$ are the Rademacher complexities[26] of $\mathcal{F}$ for the sampling of size $n_s$ from $P(x, s \neq s_{none})$, the sampling of size $n_{none}$ from $P(x, s = s_{none})$ and the sampling of size $n_c$ from $P(x)$.

The proof is provided in Appendix. Based on the Lemma 4, we can obtain the estimation error bound as follows.

**Theorem 5.** For any $\delta > 0$, with the probability at least $1 - \delta$,

$$R_{CL}(\hat{\mathbf{f}}_{CL}) - \min_{\mathbf{f}\in\mathcal{F}} R_{CL}(\mathbf{f})$$
$$\leq 4L_\phi\mathfrak{R}_{n_s}(\mathcal{F}) + 4L_\phi\mathfrak{R}_{n_{none}}(\mathcal{F}) + 4L_\phi\mathfrak{R}_{n_c}(\mathcal{F})$$
$$+ 4\frac{C_\phi K}{L}\sqrt{\frac{\ln(2/\delta)}{2n_s}} + 4\frac{C_\phi(K-L)}{L}\sqrt{\frac{\ln(2/\delta)}{2n_{none}}} \tag{15}$$
$$+ 4\frac{C_\phi K}{L}\sqrt{\frac{\ln(2/\delta)}{2n_c}}$$

where $\hat{\mathbf{f}}_{CL}$ is trained by minimizing the classification risk $R_{CL}$.

The proof is provided in Appendix. Lemma 4 and Theorem 5 demonstrate that as the number of concealed labels data increases, the estimation error of the trained classifiers decreases. This implies that the proposed method is consistent. When deep network hypothesis set $\mathcal{F}$ is fixed and $\mathfrak{R}_n(\mathcal{F}) \leq C_\mathcal{F}/\sqrt{n}$, we have $\mathfrak{R}_{n_s}(\mathcal{F}) = O(1/\sqrt{n_s})$, $\mathfrak{R}_{n_{none}}(\mathcal{F}) = O(1/\sqrt{n_{none}})$ and $\mathfrak{R}_{n_c}(\mathcal{F}) = O(1/\sqrt{n_c})$, then

$$n_s, n_{none}, n_c \to \infty \implies R_{CL}(\hat{\mathbf{f}}_{CL}) - \min_{\mathbf{f}\in\mathcal{F}} R_{CL}(\mathbf{f}) \to 0$$

Lemma 4 and Theorem 5 theoretically justify the effective of our method for learning from concealed labels. Besides, it is worth noting that this error bound is relate to the the number of classes $K$ and sample label set $L$. Lemma 4 and Theorem 5 accord with our intuition that learning from conceal labels using unbiased risk estimator will be harder if the number of class $K$ increases or the number of sample label set $L$ decreases, which aligns well with the experimental results in section 4.4.

## 4 EXPERIMENTS

In this section, we experimentally evaluate the performance of the proposed concealed labels data learning algorithm with comparative studies against state-of-the-art multi-positive and unlabeled learning and augmented classes learning approaches. Besides, we examine the issue of negative risk and perform the experiments with varying size of random sampled label set.

### 4.1 Experimental setup

**Datasets.** We employ four wide-used benchmark datasets: MNIST, Kuzushiji-MNIST, Fashion-MNIST and CIFAR-10. Additionally, we utilize two real-world concealed labels datasets, namely CLDS (Concealed Labels Data of Smoking) and CLDD (Concealed Labels Data of Disease). For CLDS and CLDD, each instance is a $156 \times 156 \times 3$ image. We report the brief descriptions of all used datasets and corresponding base model in Table.1. During training, we only use concealed labels data, which can be generated by the assumption of Eq.(3).

We collect the dataset CLDS for evaluating the effectiveness of the proposed method. CLDD is a daily scene classification dataset consists of 1350 training images and 200 testing images across 4 classes: smoking, drinking, cycling and phoning. As mentioned in the introduction, the smoking label is considered sensitive for some individuals, who hesitate to admit to smoke in their daily lives. Therefore, we generate concealed labels data by designating smoking as the concealed label in this dataset.

**Table 1: The statistics of the experimental datasets. Fashion is Fashion-MNIST and Kuzushi is Kuzushi-MNIST 5-C and 2-F NN denotes the neural networks with 5 convolutional layers and 2 fully-connected layers.**

| Name | # Training | # Testing | # Dim | # Classes | Model |
|---|---|---|---|---|---|
| MNIST | 60K | 10K | 784 | 10 | Linear, 5-C and 2-F NN |
| Fashion | 60K | 10K | 784 | 10 | Linear, 5-C and 2-F NN |
| Kuzushi | 60K | 10K | 784 | 10 | 5-C and 2-F NN |
| CIFAR-10 | 50K | 10K | 2048 | 10 | 5-C and 2-F NN |
| CLDS | 1350 | 150 | 73008 | 4 | 5-C and 2-F NN |
| CLDD | 4080 | 1020 | 73008 | 3 | 5-C and 2-F NN |

**Table 2: Classification accuracy of each algorithm on MNIST and Fashion-MNIST. $cl$ denotes the label that needs to be concealed. We report the mean and standard deviation of results over 5 trials. The best method is shown in bold (under 5% t-test).**

| Dataset | $cl$ | MPU | AREA | CoMPU | EULAC | CLCE | CLF |
|---|---|---|---|---|---|---|---|
| | 1 | 94.39 ± 0.78 | 94.52 ± 0.69 | 95.53 ± 0.62 | 69.67 ± 4.05 | 96.60 ± 0.06 | **97.72 ± 0.06** |
| | 3 | 93.85 ± 0.67 | 93.21 ± 1.12 | 95.37 ± 0.47 | 69.54 ± 2.01 | 95.82 ± 0.51 | **97.40 ± 0.32** |
| MNIST | 5 | 93.79 ± 0.09 | 93.04 ± 1.04 | 95.27 ± 0.22 | 70.53 ± 3.01 | 95.88 ± 0.15 | **97.36 ± 0.20** |
| | 7 | 94.28 ± 0.06 | 93.30 ± 0.30 | 95.41 ± 0.46 | 71.75 ± 6.38 | 95.87 ± 0.35 | **97.30 ± 0.55** |
| | 9 | 94.15 ± 0.48 | 93.45 ± 0.78 | 95.57 ± 0.43 | 63.73 ± 6.10 | 96.20 ± 0.44 | **97.43 ± 0.02** |
| | 0 | 79.20 ± 0.44 | 79.96 ± 1.50 | 80.49 ± 0.39 | 62.68 ± 1.77 | 82.22 ± 0.48 | **84.39 ± 0.55** |
| | 2 | 79.51 ± 0.54 | 79.75 ± 0.51 | 80.66 ± 1.06 | 63.63 ± 2.00 | 81.82 ± 0.54 | **83.42 ± 0.69** |
| Fashion | 4 | 79.77 ± 0.50 | 79.27 ± 1.23 | 80.58 ± 0.57 | 66.14 ± 1.30 | 81.68 ± 0.45 | **83.70 ± 0.28** |
| | 6 | 80.17 ± 0.92 | 79.31 ± 0.46 | 81.14 ± 0.68 | 63.27 ± 2.06 | 82.06 ± 0.49 | **83.01 ± 0.59** |
| | 8 | 79.84 ± 1.60 | 78.95 ± 1.53 | 81.47 ± 0.52 | 64.95 ± 1.89 | 82.91 ± 0.26 | **84.89 ± 0.12** |

**Table 3: Classification accuracy of each algorithm on Kuzushiji-MNIST and CIFAR-10. $cl$ denotes the label that needs to be concealed. We report the mean and standard deviation of results over 5 trials. The best method is shown in bold (under 5% t-test).**

| Dataset | $cl$ | MPU | AREA | CoMPU | EULAC | CLCE | CLF |
|---|---|---|---|---|---|---|---|
| | 0 | 70.64 ± 3.02 | 76.29 ± 0.84 | 75.22 ± 1.17 | 46.09 ± 4.83 | 77.13 ± 0.04 | **81.74 ± 1.18** |
| | 2 | 67.57 ± 2.97 | 75.90 ± 4.44 | 72.63 ± 3.53 | 41.34 ± 2.76 | 77.47 ± 1.16 | **82.13 ± 0.05** |
| Kuzushiji | 4 | 67.18 ± 5.89 | 76.36 ± 1.32 | 74.54 ± 1.14 | 40.95 ± 1.93 | 77.51 ± 1.22 | **82.37 ± 0.50** |
| | 6 | 67.00 ± 2.41 | 74.79 ± 1.21 | 73.61 ± 1.67 | 48.04 ± 1.84 | 78.23 ± 0.88 | **82.85 ± 0.95** |
| | 8 | 64.33 ± 2.76 | 73.64 ± 1.45 | 73.38 ± 1.24 | 42.76 ± 6.27 | 78.08 ± 1.43 | **82.60 ± 1.51** |
| | 1 | 49.70 ± 3.02 | 54.55 ± 2.97 | 57.22 ± 1.69 | 45.22 ± 1.34 | 70.65 ± 0.80 | **71.32 ± 0.27** |
| | 3 | 50.27 ± 2.07 | 53.42 ± 2.06 | 52.62 ± 1.04 | 44.87 ± 2.35 | **70.47 ± 0.08** | 70.04 ± 0.48 |
| CIFAR-10 | 5 | 49.68 ± 1.39 | 52.24 ± 0.17 | 55.15 ± 0.24 | 45.38 ± 1.40 | 70.45 ± 0.41 | **71.02 ± 0.23** |
| | 7 | 48.46 ± 1.08 | 53.62 ± 2.81 | 55.31 ± 2.50 | 42.56 ± 1.89 | 70.31 ± 0.40 | **71.31 ± 0.45** |
| | 9 | 51.25 ± 2.14 | 56.45 ± 1.55 | 57.32 ± 1.56 | 44.44 ± 2.98 | 70.49 ± 0.41 | **71.28 ± 0.27** |

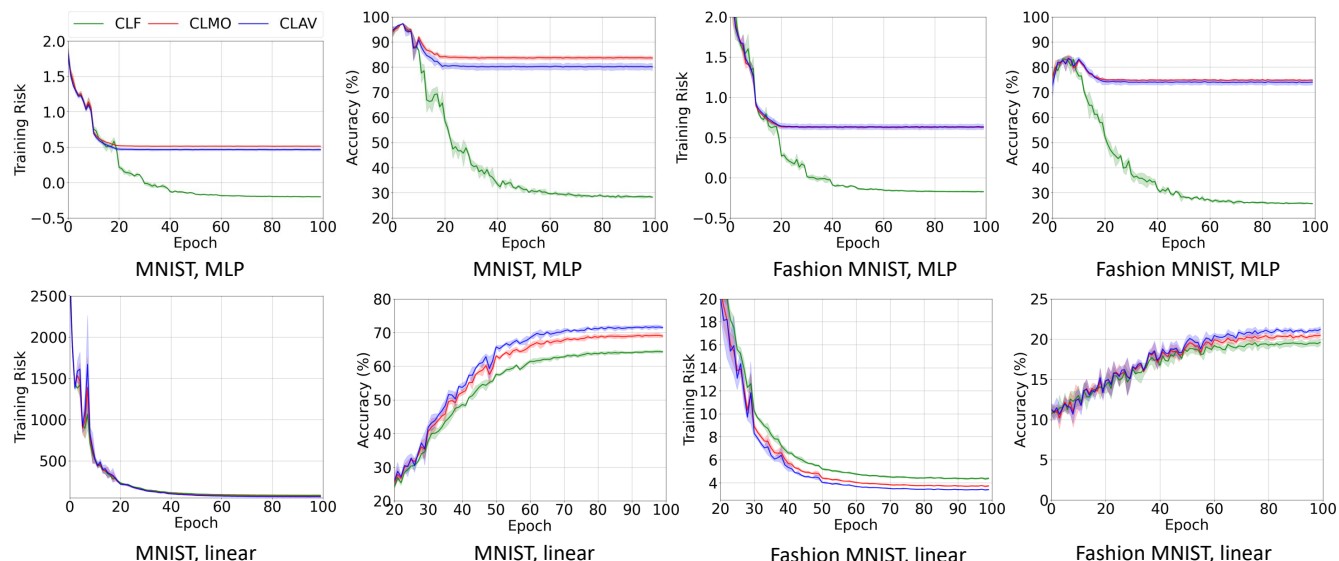

**Figure 2: Illustrations the negative risk of base models in experiments with various two datasets.**

**Table 4: Classification accuracy of different sizes of random sampled label set on Kuzushiji-MNIST.** $cl$ **denotes the label that needs to be concealed.** $L = 2$ **denotes the size of labels in the random sampled label set. We report the mean and standard deviation of results over 5 trials. The best method is shown in bold (under 5% t-test).**

| Dataset | $cl$ | $L = 2$ | $L = 3$ | $L = 4$ | $L = 5$ | $L = 6$ | $L = 7$ |
|---|---|---|---|---|---|---|---|
| | 1 | $85.85 \pm 0.11$ | $87.62 \pm 0.92$ | $89.24 \pm 0.31$ | $89.77 \pm 0.43$ | $90.23 \pm 0.35$ | $\mathbf{90.28 \pm 0.26}$ |
| | 3 | $85.61 \pm 0.69$ | $88.13 \pm 0.97$ | $89.33 \pm 0.57$ | $89.97 \pm 0.34$ | $90.50 \pm 0.34$ | $\mathbf{90.58 \pm 0.23}$ |
| **Kuzushiji** | 5 | $86.23 \pm 0.56$ | $88.41 \pm 1.10$ | $89.43 \pm 0.32$ | $90.10 \pm 0.35$ | $90.43 \pm 0.24$ | $\mathbf{90.54 \pm 0.14}$ |
| | 7 | $85.21 \pm 0.84$ | $86.95 \pm 0.45$ | $87.94 \pm 0.63$ | $88.35 \pm 0.20$ | $88.66 \pm 0.59$ | $\mathbf{89.41 \pm 0.29}$ |
| | 9 | $86.18 \pm 0.27$ | $88.24 \pm 0.48$ | $89.05 \pm 0.52$ | $89.83 \pm 0.18$ | $90.22 \pm 0.19$ | $\mathbf{90.41 \pm 0.23}$ |

**Table 5: Classification accuracy of each algorithm on real-world concealed labels datasets.** $L$ **means the number of labels in random sampled label set. We report the mean and standard deviation of results over 3 trials. The best method is shown in bold (under 5% t-test).**

| Dataset | L | MPU | AREA | CoMPU | EULAC | CLMO | CLF | CLAV |
|---|---|---|---|---|---|---|---|---|
| **CLDS** | 1 | $65.00 \pm 1.32$ | $59.33 \pm 1.52$ | $62.83 \pm 3.32$ | $50.33 \pm 2.51$ | $72.00 \pm 2.50$ | $\mathbf{72.16 \pm 1.04}$ | $71.33 \pm 2.08$ |
| | 2 | $66.83 \pm 0.28$ | $67.33 \pm 1.44$ | $63.33 \pm 2.92$ | $59.66 \pm 2.25$ | $74.66 \pm 0.28$ | $\mathbf{76.33 \pm 1.25}$ | $74.83 \pm 2.02$ |
| **CLDD** | 1 | $36.56 \pm 3.91$ | $48.22 \pm 1.37$ | $40.26 \pm 3.04$ | $40.98 \pm 1.32$ | $47.23 \pm 0.58$ | $39.71 \pm 2.81$ | $\mathbf{49.97 \pm 0.73}$ |

In addition, we also collect another real-world concealed labels dataset, CLDD, which consists of 3 classes: including normal, benign and disease. This dataset includes 1360 training images and 340 testing images for each class. In the context of data collection, the information regarding diseases of patients holds commercial value for pharmaceutical companies. Therefore, the label of disease needs to be protected during the annotation process. In this dataset, we

specifically selected disease as the concealed label for annotating examples.

**Approaches.** We compare with the 4 approaches that we have proposed in section 3, including CLF (Free, Square Loss), CLMO (Max Operator, Square Loss), CLAV (Absolute Value, Square Loss) and CLCE (Free, Cross Entropy). Additionally, we also compare with three the-state-of-the-art multi-positive and unlabeled learning

approaches MPU[38], AREA [30] and CoMPU[46], and a augmented classes learning approach EULAC [44].

For MPU, AREA and CoMPU, we treat unconcealed labels as positive labels and concealed label as the negative label. In order to fairly compare with those approaches, we also assume the class priors are known in the training stage, which play a crucial role in rewriting the multi-class classification risk. For EULAC, we treat unconcealed labels as known labels, and concealed label as unknown label. During the training stage, we assume that the mixture proportions are known to facilitate the practical implementation of the method.

To ensure a fair comparison, we adopt the same base model for all the compared approaches. We implement all approaches using PyTorch on a single NVIDIA 4090 GPU, and use Adam optimization method with learning rate candidates $\{5e-1, 3e-1, 1e-1, 8e-2, 5e-2, 3e-2, 1e-2\}$ and the weight-decay is fixed at 0.5. The mini-batch size is set to 256 and the number of epoch is fixed at 100. The hyperparameters for all compared approaches are selected to maximize the accuracy on a validation set, i.e., 10% of training concealed labels dataset.

## 4.2 Experimental results

**Benchmark datasets.** We conduct experiment on four benchmark datasets with $L = 1$ for MNIST, Kuzushiji-MNIST, Fashion-MNIST and $L = 8$ for CIFAR-10.

Table 2 and 3 show the mean and standard deviation of the test accuracy over 5 trials for different approaches. Firstly, we observe that the proposed methods consistently outperform the compared methods on all datasets, although MPU methods utilize information from the distribution of privacy labels. This demonstrates the effectiveness of our approach for learning from concealed labels. Next, from the two tables, we can see that CLF also achieve the best performance among all the approach, which uses square loss. On the other hand, CLCE achieves comparable performance, although it is slightly inferior to CLF. This can be attributed to the difference in their loss functions.

**Real-world concealed labels datasets.** Table 5 shows the mean and standard deviation of test accuracy over 5 trials for different approaches. From the table, we observe that on the CLDS dataset, the performance of CLF is better than the proposed corrected risk methods CLAV and CLMO, which aim to alleviate overfitting due to negative risk. This suggests that the deep model used in our experiments is suitable for training classifiers on this dataset, and thus the correction function is not necessary in this case. Additionally, we observe that the absolute value correction function (CLAV) performs better than the max-operator correction function (CLMO), indicating that considering both positive and negative risk contributions leads to improved performance.

On the other hand, we observe that CLAV achieves the best performance among all the approaches on the CLDD dataset. This suggests that the absolute value correction function is effective in addressing the negative risk issue and improving the classification accuracy on this dataset.

## 4.3 Issue of negative risk

We present the training risk and testing accuracy to illustrate the issue of the empirical estimator going negative when using complex models with $L = 1$. Figure 2 provides a visual representation of these results. This confirms the discussion in Section 3.3, highlighting the effectiveness of the correction function in improving the performance of the classifiers. For linear models, CLF, CLMO and CLAV have the similar performance. However, for deep models, correction functions obtain better performance than unbiased risk estimator.

It is easy to observe from the experimental results that when the risk of training becomes negative, the accuracy of classification deteriorates, especially when using complex models. However, for simple models, such experimental results do not occur. The reason behind this is that simple models typically have fewer parameters and less complexity compared to complex models.

## 4.4 Size of random sampled label set

We explore the impact of the size of the random sampled label set on the performance of the classifier. We vary the size from 2 to 7 on the Kuzushiji-MNIST dataset, and the experimental results are presented in Table 4. From the table, we can observe that as the size of the random sampled label set increases, the classifier achieves better performance. This confirms the intuitive expectation that the classifier achieves higher classification accuracy when more instances are annotated with the true label.

An important observation is that when the number of random sampled label set exceeds five, further increasing the number of instances does not improve the accuracy of the model. A reason is that the increase in labeled instances is very small compared to the existing labeled instances. Additionally, there has been a reduction in the number of unlabeled instances.

In this scenario, when the increase in labeled instances quantity is relatively small, it may have a limited impact on the model's learning capability and generalization ability. When the labeled instances quantity is low, each additional instance brings a relatively large increase in information, helping the model better learn patterns and relationships in the data. However, when there is already a large number of labeled instances, the impact of adding more instances may become smaller and may not significantly improve the model's performance.

## 5 CONCLUSION

In this paper, we introduced a novel weakly supervised learning setting and approach for learning from concealed labels. This setting is particularly useful for tasks where sensitive labels cannot be accessed during data collection. We proposed an unbiased risk estimator based on concealed labels data and improved its performance by incorporating a risk correction function. Besides, the consistency of the minimizers of proposed risk estimator is proved. The experimental results on benchmark datasets as well as real-world concealed labels datasets showed the effectiveness of our approach in various scenarios.

In the future, it would be intriguing to explore concealed labels in multi-label learning, a more challenging task compared to the problem settings examined in this paper. Additionally, another future research direction involves designing more effective methods to further enhance experimental performance.

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
