# OpenReview forum: "Learning from Concealed Labels"
_acmmm.org/ACMMM/2024/Conference — MM2024 Poster_

### Official Review · Reviewer_wrUD · 2024-05-17

**Rating:** 3
**Confidence:** 3

**Summary:**

In this paper, the authors propose a new setting named concealed labels to preserve the privacy of data.

**Strengths:**

- The idea is interesting and the proposed method is technical.
- The authors provide theoretical analysis and sufficient experimental results to verify the idea.

**Limitations:**

- The proposed setting seems to be a combination of partial label learning and open-set learning. If we regard NONE as a class, the setting is identical to partial label learning.

- The theoretical analysis seems to be inspired by the analysis of partial label learning.

**Suitability:**

2

---

### Official Review · Reviewer_zESG · 2024-05-19

**Rating:** 3
**Confidence:** 2

**Summary:**

This paper presents a concept for protecting sensitive information classification tasks via the notion of "concealed labels". This method prevents sensitive labels from appearing during the data collection stage. The authors prove that minimizing classification risk with concealed labels can provably lead to the minimization of standard classification risk, which can then subsequently lead to estimation error bound based on Rademacher Complexity. The framework is validated through experiments on synthetic and real-world datasets, demonstrating its effectiveness in classifying instances from both insensitive and sensitive labels while ensuring privacy protection.

**Strengths:**

1. This paper proposed a new type of weakly-supervised learning paradigm, which is known as learning with Concealed Labels (LCL), which plays a similar function as Label Proportion Learning (LPL) - it is guaranteed to be learnable while does not disclose sensitive label information.

2. LCL is seemingly better, as it requires less constraint compared with Label Proportion Learning and other alternatives.

3. Authors proved that LCL can result in unbiased risk minimizer as conventional supervised learning, following established proof structure [1].

4. LCL also exhibits superior performances compared to its counterparts on commonly used benchmarks.

[1] Classification from pairwise similarity and unlabeled data, ICML 2018.

**Limitations:**

1. I feel the concept proposed in this paper is very hard to follow; I cannot get the gist of the "concealed label". If I understand correctly, what the authors are proposing is that instead of giving sensitive labels for users to answer to, the "concealed label" defines a set of insensitive labels, where they are collectively mapped to a sensitive label.

2. Can authors provide a proof sketch on why Equ. 4 holds? It is not evident to me why supervised risk can be expressed in terms of concealed label. If Equ 4. does not hold, the correctness of all subsequent theorems cannot be guaranteed.

3. The scenario setup in this paper is somewhat confusing - are we trying to avoid asking the users to provide sensitive information, or are we trying to avoid disclosing this information to an adversary?

**Suitability:**

2

---

### Official Review · Reviewer_cqtm · 2024-05-22

**Rating:** 4
**Confidence:** 2

**Summary:**

sampled insensitive labels as concealed labels set to mask the sensitive data. Additionally, an unbiased estimator is constructed from the concealed data, with its estimation error bounds being derived. Experimental results confirm the effectiveness of the proposed method in handling concealed labels across both synthetic and real-world data sets.

**Strengths:**

(1)	This paper introduce a novel concealed label strategy to protect privacy of each instance.
(2)	This paper provides theoretical support for the method, including an unbiased estimator for empirical risk and the corresponding estimation error bound. Besides, two correction empirical risk using max-operator and absolute-operator is given for enforcing the classification risk to be non-negative.

**Limitations:**

(1) This paper requires a clearer expression of the method and innovative points in introduction section.
(2) The definition of M=P(x) is not given in the next line of formula (4) on page 3.
(3) The expression following the second equals sign in Equation (8) is incorrect on page 4.
(4) In the experiments, there is no comparison with sufficient comparison methods and recent works, which is difficult to prove the effectiveness of your method at present.
(5) From the observations in Figure 2, it is noted that the performance of the CLF method on both the MNIST and Fashion MNIST datasets, regardless of whether a linear model or a Neural Language Processing (NLP) model is employed, does not achieve the accuracy levels indicated in Table 2.

**Suitability:**

2

---

### Meta-Review · Area_Chair_VmvX · 2024-07-02

**Recommendation:** Accept (Poster)
**Confidence:** 5

**Metareview:**

Annotating data for sensitive labels (e.g., disease, smoking) poses potential threats to individual privacy in many real-world scenarios. To address this issue, a novel setting is proposed to protect the privacy of each instance, termed learning from concealed labels for multi-class classification. Concealed labels prevent sensitive labels from appearing in the label set during the label collection stage, as illustrated in Figure 1. This figure specifies none and some randomly sampled insensitive labels as concealed labels to annotate sensitive data.

In this context, an unbiased estimator can be established from concealed data under mild assumptions. The learned multi-class classifier can not only accurately classify instances from insensitive labels but also recognize instances from the sensitive labels. Additionally, the estimation error is bounded, demonstrating that the multi-class classifier achieves the optimal parametric convergence rate. Experiments validate the significance and effectiveness of the proposed method for concealed labels in both synthetic and real-world datasets.

This paper considers a very interesting problem setting and sounds reasonable to protect privacy. Actually, in CoLL, there is already a paper to discuss that it can be used to protect privacy. And this paper formalizes it well.